# The Wildman Programme. A Nature-Based Rehabilitation Programme Enhancing Quality of Life for Men on Long-Term Sick Leave: Study Protocol for a Matched Controlled Study In Denmark

**DOI:** 10.3390/ijerph17103368

**Published:** 2020-05-12

**Authors:** Simon Høegmark, Tonny Elmose Andersen, Patrik Grahn, Kirsten Kaya Roessler

**Affiliations:** 1Department of Psychology, University of Southern Denmark, 5230 Odense M, Denmark; tandersen@health.sdu.dk (T.E.A.); kroessler@health.sdu.dk (K.K.R.); 2Department of Work Science, Business Economics and Environmental Psychology, Swedish University of Agricultural Sciences, SE-230 53 Alnarp, Sweden; patrik.grahn@slu.se

**Keywords:** nature-based intervention, men, stress, quality of life, chronic illness, mental health, nature–body–mind–community (NBMC), biophilia, supportive environment theory (SET)

## Abstract

Many men have poor mental health and need help to recover. However, designing a rehabilitation intervention that appeals to men is challenging. This study protocol aims to describe the ‘Wildman Programme’, which will be a nature-based rehabilitation programme for men on long-term sick leave due to health problems such as stress, anxiety, depression, post-cancer and chronic cancer, chronic obstructive pulmonary disease (COPD), cardiovascular disease, or diabetes type II. The programme will be a nature-based rehabilitation initiative combining nature experiences, attention training, body awareness training, and supporting community spirit. The aim of the study will be to examine whether the ‘Wildman Programme’ can help to increase quality of life and reduce stress among men with health problems compared to treatment as usual. The study will be a matched control study where an intervention group (number of respondents, *N* = 52) participating in a 12-week nature-based intervention will be compared to a control group (*N* = 52) receiving treatment as usual. Outcomes are measured at baseline (T1), post-treatment (T2), and at follow up 6 months post-intervention (T3). The results of this study will be important to state whether the method in the ‘Wildman Programme’ can be implemented as a rehabilitation offer in the Danish Healthcare System to help men with different health problems.

## 1. Introduction

The prevalence of people in Denmark experiencing high levels of stress has increased from 20.8% in 2010 to 25.1% in 2017 [1]. Poor mental health and other mental disorders are also increasing problems in the Danish population.

Far more women than men experiencing stress and stress-related diseases seek help in the Danish healthcare system [2,3]. A possible explanation is that the Danish healthcare system lacks attractive rehabilitation interventions for men [3]. Another possible barrier can be the view of men in the Danish culture. Many men feel they are not allowed to be vulnerable, and therefore, it can be difficult for them to reach out for help when they go through a life crisis [2,4].

### 1.1. Rehabilitation In Natural Environments

Since ancient times, there has been a belief that certain natural areas possess special qualities that can bestow delectation and health for a troubled soul. Already in the Gilgamesh epic, written almost 5000 years ago, it is told how King Gilgamesh could find consolation, strength, and health in a magnificent garden [5]. In sacred writings such as the Bible and the Quran, in the poems of the Roman author Vergilius and in countless poems, novels, and songs thereafter, the longing for pleasant natural environments has been expressed. However, throughout this time, there has also been descriptions of unpleasant and unhealthy natural areas, where some even provoke horror that people shun. Divina Commedia by Dante Alighieri (1320) [6] describes Hell in images of stinging insects, storms, swamps, iciness, darkness, red hot deserts, and choking gases. Research has shown that humans reflexively avoid certain places and situations [7,8], while others are attractive [9,10].

The word “nature” derives from the Latin “natura”, which literally means “birth”. Early on, the word gained a broad significance, in being the basis or inherent quality of individual things or persons as well as life on earth or the universe [11]. The word “culture” derives from Latin’s “cultura”, which means “cultivate”, but also figuratively “care and honoring”, where “cultura” originates from Latin’s “colore” which means “to tend” and “guard” [11]. Thomas Beck [12] writes about the two Renaissance philosophers Jacopo Bonfadio and Bartolomeo Taegio. Both published in the mid-16th century thoughts on how nature should be described and interpreted. They defined the wilderness as the first nature, the cultural landscape with fields and pastures as the second nature, and the garden as the third nature. Beck claims that their thoughts were not new but had already been described in ancient Rome by, for example, Pliny the Elder, Lucretius, and Cicero. The third nature represents the finest interaction between nature and culture, not least art and nature, according to Bonfadio and Taegio [12]. In landscape architecture, this definition of nature has been strong. Today, especially the first nature is seen as threatened, but also the second and third nature. This has led to talk of a fourth nature, to recreate environments with high biodiversity, both in urban areas and in rural areas [13]. Kronlid writes on the view of the first nature that a survey among landscape architects shows that it can be divided into three parts: the “free nature”, completely free to develop under its own conditions without being contaminated by visible human influence; the “non-free nature” that is used to satisfy needs in, for example, tourism and recreation as well as a type of “false nature”, a false construct, completely subordinate to human influence as a production landscape of, for example, timber and other raw materials [14]. This tradition is also strong in environmental psychology, and not least where environmental psychology, landscape architecture, and nature’s impact on health are combined; e.g., when it comes to nature-based rehabilitation. Terry Hartig and colleagues write the following: ”We are particularly concerned with nature as the seemingly natural features and processes that people ordinarily can perceive without the use of specialised instruments or sensory aids. (…) the terms ‘nature’ and ‘natural environment’ get used somewhat interchangeably, although, in a seeming contradiction, the nature of interest here is not only found in natural environments, but also in otherwise built environments. (…) People may enjoy urban parks, botanical gardens, and golf courses as representations of natural environments, while still knowing of their artificial character” [15]. More research is exploring the health promoting qualities in nature and which qualities have most effect regarding e.g., age, gender, or health; at work or leisure; in schools, hospitals, at home, etc. One of the ways to describe these qualities is by the Perceived Sensory Dimensions (PSD) [16].

Previous research suggests that spending time in natural environments and using nature therapy in rehabilitation programmes can increase quality of life, reduce symptoms of heart disease [17,18], relieve symptoms of stress [18,19,20,21,22], reduce symptoms of anxiety and depression [23], and increase the ability to return to work after sick leave [24]. Most of these studies have been conducted in more urban green areas and gardens. However, studies have also been conducted in forest areas outside built-up areas. A large number of scientific studies suggest that forest therapy is a beneficial intervention in reducing stress levels and depression in adults. However, most studies are not sufficiently well accomplished regarding control groups and accuracy [25,26].

Given that natural environments cause different emotions, how come humans do recover from high levels of stress and mental fatigue in these environments [27]? Research show that people seem to have an innate ability to feel danger or security in natural environments. Two theories start from this evolutionary perspective: The Stress Reduction Theory (SRT) [28,29] and the Attention Restoration Theory (ART) [30]. Both of these theories are developed to explain why, for example, walking in urban green areas restores humans from high levels of stress and mental fatigue.

SRT was developed by the American environmental psychologist Roger Ulrich [28]. Ulrich argues that despite most of us live in urban environments, the human genome is identical to our ancestors. Back in more primitive ages, man was subject to the terms of nature and forced to trust his basic emotions, i.e., his unconscious and instinctive bodily reactions. Today, in urban environments, we are bombarded with information most of the time, and our basic emotions are not adapted to these new living conditions. This can cause affective hyperactivity and difficulties with emotional regulation that can lead to stress, anxiety, and depression. In natural environments, compared to urban environments, it is easier for us to judge when a threat is over, which makes it possible for our body and mind to release the sympathetic stress response after a threat and subsequently relax and recover. However, it is important that the natural environments are experienced without threats when it comes to recovery and healing. Nature without threats automatically causes positive feelings and a parasympathetic physiological response that is associated with feelings of calmness, relaxation, comfort, and fascination [28,31].

ART was developed by the two psychologists Stephen Kaplan and Rachel Kaplan and describes attention processes that take place in nature. These processes occur on a perceptual cognitive level and have a recovering effect. The theory describes the healing effect of nature and defines two different kinds of attention: (1) directed attention and (2) soft fascination. We switch between the two kinds of attention during our everyday life according to our present tasks and environment [32,33]. The ART theory is based on the assumption that people have two systems of attention: directed attention and fascination. Directed attention is used when we work in a goal-oriented way and need to inhibit distractors, i.e., when driving a car or working in an office. This requires a huge amount of energy and is an exhaustive resource involved in executive processes in thinking and may lead to directed attention fatigue [34]. Fascination catches our attention involuntary, and we do not need to or can decide when to use it. Fascination can be hard, i.e., what catches the attention leaves no room for reflection. For example, it may be that you are watching an exciting competition in sports. However, in nature, most of the things you experience are of a soft character. You may see a gleam of water or hear a bird singing. It does not require tiring decisions; instead, you can rest in the experience. This increases our mental capacity and has a restoring and recovering effect on our body and mind [32]. According to ART, our senses are stimulated in nature and here, we automatically use the soft fascination, which means we can relax and rest. The theory has been tested in many studies with encouraging results [35,36]. However, there are criticisms of some assumptions, such as the evolutionary basis of the mechanism and the definition of soft fascination [37].

Both theories, SRT and ART, are based on evolutionary theory. They claim that not all natural areas are restorative. The restorative natural environments should contain special characteristics that most people interpret as safe and restful, and the theories involve a number of such qualities [29,32].

A theory based on nature-based interventions with staff and which has the potential to be developed further is the Supportive Environment Theory (SET) [27,38,39]. The theory incorporates the theories SRT and ART and deals with synergies between the capacity of natural areas to heal people and the professional knowledge of the staff. SET was developed by the Swedish Professor and landscape architect Patrik Grahn, and it focuses specifically on the treatment of stress-related mental illnesses in therapy gardens [24,40]. SET describes which qualities of natural environments we prefer and where we feel safe [16]. In these safe environments, we are able to relax and recover. SET is a biopsychosocial theory [41] based on the evolutionary assumption that human beings have evolved in close contact with natural environments living in small social and cultural communities. Since we have evolved this way, we find some environments inherently more supportive than others. Supportive environments have physical (both built and natural environments), social (e.g., relatives, friends, or colleagues), and cultural (e.g., values, historical context, traditions) qualities. According to SET, we need supportive environments to develop physically and mentally [38]. Research on long-term nature-based supportive environmental rehabilitation programmes (up to a duration of 24 weeks) shows better results and a reduction in symptoms of stress and depression in the participants and an increase in the participants’ return to work [24]. SET proposes the development of health-promoting synergies regarding support from the three different spheres (the social, cultural, and physical environment) through activities led by a professional staff.

In this context, the following are of interest to highlight: Research shows that meditation and breathing exercises can activate the parasympathetic nervous system and help reduce symptoms of stress [42] and depression [43,44], and they can be used in nature-based therapy with positive results for health and well-being [45,46,47] In addition, a systematic review shows that combining breathing exercises and body awareness training such as Qigong can enhance mental health [48,49]. Research in the field of Qigong shows that higher levels of energy and better body awareness can have a positive impact on the participants’ mental and physical health [49].

Furthermore, ecotherapy [50] can help strengthen the social sphere; the community feeling [51] and being part of a social community with others can enhance feelings of meaning and life purpose [52]. The creation of a common meaning and a sense of coherence in the group can improve the participants’ health and quality of life [52]. Ecotherapy suggests that many people today need to rebuild their contact with nature and that sensory experiences in nature can activate the parasympathetic nervous system and help to restore the body and mind. For example, focusing on the different sensory experiences in nature, such as scents, sounds, tactile stimuli of nature objects and focus on details as well as the wholeness of nature’s diversity gives a sense of well-being [50,51]. According to SRT, it is easier for people to interpret and value sensory experiences from nature [28,29].

### 1.2. The Wildman Programme

A Swedish study shows that the interest in outdoor life among people in Sweden is great in relation to forests, beaches, the sea, meadows and hills, regardless of age and ethnicity. There are, of course, differences between the groups when it comes to outdoor life. For example, people aged 45–64 visit natural environments more often than those who are 25–44 years old; those who live in the countryside visit natural environments more often than people living in the cities; and people born outside Europe visit natural environments less often than those born in Europe. The interest in outdoor life is nevertheless great. For example, 57.2% of those born outside Europe visit natural environments frequently or very often, while the corresponding figures for those born in Europe are 67.5%. We must assume that it is the same in Denmark [53]. Nature-based interventions in gardens, such as horticultural therapy, mostly attract women [24]. In various studies, men have shown a strong interest in the wilderness and activities in the wilderness, including multi-day trips with overnight stays [54]. A North American study found that the difference between various ages and ethnic groups among men was relatively small [55]. Interest has been relatively constant since the 1980s [56], and it is not the lowest among men in northern Europe [57]. The value lies in the challenges being concrete and perceived as meaningful. If they are manageable, the participants experience a reward; for example, it may be about coming to a place experienced as beautiful and breath-taking [58]. In an effort to offer a rehabilitation programme that appeals more to men, we developed the ‘Wildman Programme’. The ‘Wildman Programme’ is a nature-based rehabilitation initiative (NBR) combining nature experiences in rural settings (presentation of scenic areas, sensory activities, silent walking, and fascinating stories about nature and how we are connected to the big circle of life), attention training (walking, standing, sitting and lying meditations, outdoor sittings, and narrative meditations), body awareness training (breathing exercises, outdoor playing, balance training, Qigong, and other kinds of physical activities), and supporting community spirit (bonfire cooking, talks, and storytelling around the bonfire). The overall goal of the programme is to improve quality of life and reduce symptoms of stress among the participants. The ‘Wildman Programme’ is designed to appeal to men, and especially to men who do not feel motivated to participate in other rehabilitation programs. If this program proves to work, it can broaden the range of rehabilitation measures for men. The approach of the programme is inspired by the SET theory, including ART, SRT, ecotherapy, and activities such as meditation and Qigong. In addition, the men are introduced to knowledge from evolutionary theories, which explain how still today in modern society we have to rely on the instincts and basic emotions we had when we lived as hunters and gatherers in nature many thousands of years ago.

We believe that one of the problems in modern Western societies today is that we have disconnected ourselves from nature, and that makes us sick. Rebuilding the connection with nature is important for our well-being, since we have always lived in close contact with nature. Nature connection, as we define it, includes place attachment. It is about the interaction between person, process, and place and contains both cognitive, affective, and behavioural components. The person in question must feel that the place and the activities it invites provide security, meaning, and involve a flow of self-rewarding activities that can bestow a moment of happiness [59,60,61,62]. The evolutionary explanation, which can be related to the theories above, we assume can motivate the participants in the ‘Wildman Programme’ and inspire them to feel hope for the future and get back in contact with their inner strengths, which are feelings the participants have often been missing for a very long time because of symptoms of stress and health problems. Understanding our evolutionary connection to nature and how our body and mind are adapted to live close to nature can, according to e.g., Jordan and Hints [51], reduce stress and improve quality of life.

### 1.3. Aim

The aim of the present study is to investigate the effect of a nature-based intervention ‘The Wildman Programme’ on quality of life and symptoms of stress among men with long-term health problems compared to men receiving treatment as usual. A further aim is to investigate which natural environments in rural areas work best as supportive environments in rehabilitation. The purpose of this study protocol is to inform about the study and how it is intended to be conducted. Moreover, through the review process, the design and focus of the study can be improved.

Our expectation is that nature-based rehabilitation is suitable for men with depression and stress or long-term illnesses that can lead to depression and stress, and that the nature-based ‘Wildman Programme’ accordingly can help men with a wide variety of chronic illnesses.

Thus, we expect that the participants receiving the intervention will first and foremost experience a significant improvement in their quality of life measured by the World Health Organization Quality of Life Instruments (WHOQOL-BREF) scale [63] corresponding to a 20% improvement. WHOQOL-BREF is an abbreviated generic Quality of Life Scale developed through the World Health Organization.

Furthermore, we assume significant improvements in the other secondary effect outcomes (symptoms of stress, self-experienced restitution, and return to job) compared to the participants receiving treatment as usual (TAU). As far as we know, this type of intervention or research has never been conducted before.

## 2. Materials and Methods

We choose to conduct the study as a matched-controlled study. An intervention group is receiving nature-based therapy through the ‘Wildman Programme’ and will be compared to a matched control group receiving TAU i.e., case management (see Figure 1). Participation in the ‘Wildman Programme’ is voluntary, and the study has been ethical approved by the University of Southern Denmark Research & Innovation Organization, SDU RIO.

### 2.1. Venue

The intervention programme takes place in five different natural environments in the countryside (Table 1). They are not located far from urban areas, but participants in a previous pilot study have experienced that they are completely surrounded by natural environments [64,65].

Every second time the group meets during the 12-week course, they meet up at their base camp to create a safe base for the group processes. The base camp is located close to a nature school and consists of a fireplace and a shelter. Every other time, the group meets in a new nature environment with different qualities in the southern part of Funen.

The nature environments are selected with inspiration from the Perceived Sensory Dimensions (PSD): serene nature rich in species and refuge, which are identified as most important to people with symptoms of stress [16,66,67]. The five local nature environments (see Table 1) have been selected during the pilot project [64,65]. They are based on feedback and preferences from former participants. The former pilot study shows that the participants have preferences for different types of nature; therefore, shore, forest, tunnel valley, hill, and meadowland are represented as settings in the ‘Wildman Programme’. These natural environments are carefully selected but not unusual or exceptional; rather, they represent a spectrum of natural environments in Denmark. The nature environments are also selected so that the participants easily can access them by private or public transport and the environments are easy for the participants to visit alone or with their families after the course has ended.

### 2.2. Participants

Inclusion criteria: The participants are men, aged 18–76 years old, living in the region of southern Funen in Denmark. The male participants are on long-term sick-leave and suffer from stress (ICD: F43.8 and F43.9), anxiety (ICD: F41.2), or depression (ICD: F32.0) according to the ICD, the International Classification of Diseases [68]. Moreover, the participants can suffer from diseases such as chronic cancer, post-cancer, chronic obstructive pulmonary disease (COPD), cardiovascular disease, or type 2 diabetes—all of which can cause stress. We expect that the nature-based ‘Wildman Programme’ can help men with a wide variety of chronic illnesses improve their quality of life and reduce symptoms of stress. Many of the participating men included in the study will in the beginning of the course to some extend experience lower quality of life and not have benefitted sufficiently from participation in other available rehabilitation interventions.

The participants will be divided into groups of 10–18 individuals in the ‘Wildman Programme’. Approximately 40% of the participants will be recruited from the health centre in the Municipality of Svendborg and the Municipality of Faaborg-Midtfyn, approximately 40% will be recruited from the jobcentre in the Municipality of Svendborg and the Municipality of Faaborg-Midtfyn, and the remaining 20% will be recruited from the general practitioners. The control group will be recruited as a form of a natural experiment from comparable departments in the health centre and the job centre and from other general practitioners.

The study will include at least 104 participants: 52 in the intervention group and 52 in the control group. The diagnoses, ages, job status, and social status of the participants will be recorded. Instead of selecting participants to be included in the intervention group and TAU by randomisation, we have chosen to follow two groups via a type of natural experiment or quasi-experiment. We choose to select the TAU group from matched health centres, job centres, and general practitioners via their normal standard routines, where the groups are matched with respect to age, gender, and diagnosis [69]. The matched control group (*N* = 52) of men will resemble the men in the intervention group with regard to the distribution of age, job status, and levels of stress to ensure that the participants in the intervention and control group are as similar as possible (see Figure 2).

The participants in the control group receive traditional treatment and rehabilitation in the healthcare system of the municipalities, which do not take place in nature. They participate in physiotherapeutic treatment, fitness training, mindfulness courses, and traditional rehabilitation programmes (consisting of e.g., physical exercise, nutritional counselling, and education about their illness).

The exclusion criteria are psychosis or psychotic disorders, brain injuries, addiction or physical disabilities that prevent them from participating in the physical exercise programme or move about in nature.

### 2.3. Intervention

The ‘Wildman Programme’ is led by a nature guide and a health professional (e.g., a physiotherapist, a nurse, or a psychologist) who are experienced in rehabilitation of the included diagnosis and further educated in the nature-based rehabilitation methods. Based on past experiences in the ‘Wildman Programme’, the drop-out rate is expected to be approximately 15% due to the participants’ return to work, hospitalisation, or because the intervention does not suit them.

The intervention has a duration of 12 weeks with one weekly three-hour meeting [24]. In addition, the participants are given homework such as breathing exercises and meditation techniques for around 15 min a day, and they should give themselves breaks during the week by spending time in a self-selected supportive environment in nature in their home area. There are 10–18 participating men in each group of the ‘Wildman Programme’.

The intervention has a permanent chronological structure, which is the same every time the intervention groups meet, but the intervention elements are adjusted accordingly to seasons and weather conditions.

The ‘Wildman Programme’ is described as a ‘Nature–Body–Mind–Community’ (NBMC) approach focusing on nature experiences, body awareness training, attention training, and supporting community spirit explored in the group at base camp and in five other nature environments with different nature qualities.

The NBMC approach has been developed in a pilot project during the period 2014–2018 [64,65]. The ‘Wildman Programme’ consists of the following main elements: (1) Nature environments and nature experiences, (2) Body awareness, (3) Mind relaxation and meditation, and (4) Fire talks, storytelling, and community spirit. In nature, the participants in the ‘Wildman Programme’ will experience bonfires, storytelling, meditation, breathing exercises, and sensory perceptions to reconnect with nature. They listen to the birds, the wind, and the sounds of the creek, and they feel the soil and the leaves and bark on the trees. They lay on their back on the ground, looking up in the treetops, seeing the shades and colours of light and the motion made by the wind; they lie totally quiet and just are in soft fascination [16,32,70,71].

### 2.4. Nature Environments and Nature Experiences

In the ‘Wildman Programme’, the participants will be introduced to different nature environments that are expected to have a supportive effect. The environments are selected based on nature qualities that according to the Perceived Sensory Dimensions (PSD) can make the participants feel comfortable and inspire to inner peace which, according to SET, can reduce stress [16,27]. Nature environments and nature experiences belong first and foremost to the sphere “nature” in SET.

The participants will also be supported to sense and restore their contact with nature. In the programme, they will regularly be trained in enhancing their sensory perception of nature [59,72]. Experiencing nature alternately through different senses are expected to help them be more present and forget about negative thoughts in the moment, since they are focusing on the aesthetics of nature [28,39,59,73].

In the ‘Wildman Programme’, the participants are also presented with biological knowledge and stories about nature and our evolutionary relationship with nature, which also are expected to help them move from their directed attention to soft fascination. Wondrous stories about nature (e.g., selected stories about trees, plants, animals, or other natural phenomena) can make nature more accessible to the participants and ‘open’ nature to the men in the ‘Wildman Programme’ [74]. The ‘Wildman Programme’ will take place all year round and in all kinds of weather, which gives the participants the opportunity to reconnect with the cycles of nature, the seasonal changes, and the powers of nature.

#### 2.4.1. Body Awareness

Many of the men participating in the ‘Wildman Programme’ will have little body awareness and experience problems with their balance. Therefore, in the ‘Wildman Programme’, the participants will be introduced to balance exercises and body and stretching exercises inspired by research on Qigong [48].

Calligraphy health Qigong is an approach that is easy for the participants to engage in and can easily be adapted to the different physical needs in the group and the different nature environments. Qigong exercises are expected to help the participants experience a flow of energy in the whole body and strengthen their body tone and body awareness. In addition, the ‘Wildman Programme’ includes different kinds of outdoor play. These can help the participants strengthen both their physical and mental health, since they increase their heart rate and get their blood circulation activated, while at the same time, they get to play and laugh together [75].

#### 2.4.2. Mind Relaxation And Meditation

During the ‘Wildman Programme’, the participants will also experience body scanning meditations, outdoor sittings, and narrative meditations. The practice will be adapted to the changes in seasons and the weather in Denmark and accordingly, the meditation practice will shift between sitting, standing, walking, laying, and working forms of meditation to keep the participants from getting cold. The narrative meditations are inspired by the mindfulness concept [76] and integrate nature elements and knowledge about our life in nature through time.

In the ‘Wildman Programme’, the participants will learn different breathing exercises that activate the parasympathetic nervous system. The breathing exercises will be repeated in the course and can be done by the participants at home both during and after the course. Breathing exercises are expected to help reduce stress and have a positive physiological impact on the body. Breathing exercises can enhance the sensory experience of nature and can enhance feelings of well-being and reduce stress [42].

#### 2.4.3. Fire Talks, Storytelling, and Community Spirit

An important aim during the ‘Wildman Programme’ is to create positive interactions among the participants. Being together in nature can strengthen positive relationships because social barriers and hierarchy are reduced. Therefore, the participants can experience being more open to others in the group. Research indicate that activities in natural areas often become more spontaneous, since the environment is more unaffected, far less rigid, strict, and organised than in built environments. This in turn can lead participants to be more informal in their contact with each other in natural environments, which in turn can lead to opportunities of spontaneous meetings and networks [77]. This aim relates to the “social” and “culture” spheres according to SET. The ‘Wildman programme’ focuses on creating a team spirit in the group and a common culture where it is allowed to be both vulnerable and resourceful at the same time. The participants should feel safe in each other’s company and be able to tell their own story as well as create a common story in the group [78]. In this aspect, the talks in the group around the bonfire have a significant importance. Evolutionarily, bonfires have offered a kind of refuge and safety from predators [79], and these feelings of safety and refuge can also today make it easier for the participants to be open-minded, to let go, and be present.

One of the goals of the ‘Wildman Programme’ is to create positive social interactions and a remaining network after the programme has ended. Hence, the ‘Wildman Association’ has been established as a club for former participants of the ‘Wildman Programme’, and it is driven by volunteers. On the last day of the course, the participants will meet with the ‘Wildman Association’ consisting of previous participants of the ‘Wildman Programme’, and they get the opportunity to become members of a big network that continues to meet in nature and use some of the techniques they have learned during the ‘Wildman Programme’.

### 2.5. Outcomes

The participants all experience symptoms of stress and reduced quality of life due to their physical or mental health problems. Therefore, the following outcomes have been selected. All primary and secondary outcomes are measured at baseline (T1), post-treatment (T2), and at follow up 6 months post-intervention (T3).

Primary outcome:

The primary outcome is to investigate whether the results from the NBMC approach in the ‘Wildman Programme’ have a significant effect on the participants’ quality of life. The effect will be measured by the following scale:

Self-experienced quality of life: the World Health Organization (WHO)’s brief questionnaire, WHOQOL-BREF [80].

Secondary outcomes:

The secondary outcomes are to investigate whether the ‘Wildman Programme’ has a significant effect on the participants’ level of stress, self-experienced restitution, and job status. The effects will be measured by the following methods:Level of stress: Cohen’s Perceived Stress Scale (PSS) [81].Self-experienced restitution: The Perceived Restorativeness Scale-11 (PRS-11) [82].Job status: Measured by self-reported job status before, post, and 6 months after the intervention.

## 3. Research Design

### 3.1. Flow of Participants

The referred men will be invited to an introductory interview before they can be included in the ‘Wildman Programme’. The sample size in the study will be 104 in total: 52 men will be included in the intervention group and 52 men in the control group will be given treatment as usual (see Figure 2).

The strategy to achieve adequate participants in the study is to establish close cooperation with the health centres and job centres in the two municipalities, which have decided to hire a project leader to coordinate the recruitment to the intervention and control group.

The participants in the intervention group and the control group will be answering a questionnaire at baseline (T1), post-treatment (T2), and at follow up 6 months post-intervention (T3).

The questionnaire will include the following scales:Self-experienced quality of life: WHO’s brief questionnaire, WHOQOL-BREF, which examines four domains in a five-point Likert scale: physical health, mental health, social relationships, and health-related environments, for instance access to medical care. The global quality of life is based on the participants’ scores on the four domains, and they range from 0 to 100, with a high score indicating a high quality of life [80].Level of stress: Cohen’s Perceived Stress Scale, PSS, examines the way you handle stress in daily living on 14 items in a five-point Likert scale [81].Self-experienced restitution: The Perceived Restorativeness Scale-11 (PRS-11) [82] measures with 11 items four different categories of self-experienced restitution related to spontaneous attention: fascination, being away, coherence, and scope. The scale is used for measuring meditation practice and attention training in natural environments.Job status: Measured by self-reported job status before, post, and 6 months after the intervention.

Participants from the intervention group and control group will be contacted by e-mail or by phone if they have not answered the questionnaires. No outcome data will be collected on participants that drop out of the study.

No other nature-based interventions are permitted during the trial.

Data will be securely stored in a locked electronic folder at the University, and data will be analysed in SPSS.

### 3.2. Statistical Analysis

As we are dealing with a mixed group of participants, the power calculation will be based on an estimate of the medium value (*M*) and the standard deviation (*SD*). Based on population standards for men with low measures on the WHOQOL-BREF scale, the following medium value and standard deviation are estimated: *M* = 57.0, *SD* = 18.0 [63]. According to the population standards for men [83], a 20% improvement equals a change from a low global quality of life to a high global quality of life. In the light of our previous experiences with the courses and the results obtained, a medium size effect can be expected when it comes to self-reported quality of life measured with WHOQOL-BREF (Cohens d = 0.50). A t-test power calculation made by the GPower 3.1 [84] shows that there must be a total of 104 participants to obtain a medium effect on the WHOQOL-BREF (Cohens d = 0.60), by setting the significance level to 5%, the power to 80%, and the expected drop out by the 6 months’ follow up to 15%.

Data from the pilot project showed an improvement in the quality of life for the target group in relation to the mean value and standoffs.

The results of the study will be analysed according to the intention-to-treat perspective. Primary and secondary outcomes at baseline, after end of treatment, will be analysed, and again at a six months’ follow-up applying linear mixed-effects models. Missing data will be dealt with by the maximum likelihood method, as this method is regarded as the optimal method for handling missing data caused by drop out [85].

The effect of the treatment on the global quality of life mediated by the perceived restrictiveness scale is also examined by the Preacher and Hayes macro process for SPSS. The method is based on the maximum likelihood estimation and bootstrap [86].

## 4. Discussion

Gradually, more and more research show that the use of nature in health promotion and treatment has positive effects [24,87,88,89]. In Denmark, we still do not have the same practice to integrate nature on a strategic level in health programmes as in e.g., Sweden, Norway, and the Netherlands [90,91], but more and more development and research into nature-based interventions has taken place in Denmark in recent years [92,93].

The novelty of this study is the investigation of nature-based methods adapted to *men*, who are often an overlooked group in the healthcare system. Too many men do not seek help in the healthcare system, and they reject health offers or drop out at a much bigger scale than women. Traditionally, it has been difficult to create rehabilitation programmes that appeal to men [2]; therefore, it is important to develop and implement new methods to help men. The ‘Wildman Programme’ is designed based on the knowledge of men and their needs, and the activities are moved from inside the healthcare system to natural areas, which often appeal to many men and better resonate with their self-perception. The results of this study will be important to state whether the method in the ‘Wildman Programme’ has potential to be implemented as a rehabilitation offer in the Danish healthcare system to help men with different health problems gain better life quality and reduce symptoms of stress.

A pilot project completed prior to this study showed that nature has the potential to have a special appeal to men and that most men agree that being in nature has a positive effect on their quality of life and can increase feelings of joy, power, and peace [64,65]. In addition, the novelty of the study is to investigate whether the ‘Wildman Programme’ has a structure and contains methods that are easily accessible and easy to integrate for healthcare professionals in healthcare programmes in municipalities in Denmark. More health care professionals are asking for concrete programmes and tools they can use to support participants and patients with health problems in nature, and they are asking for recommendations on specific places in nature that are particularly favourable for nature-based interventions. This study can hopefully contribute with new knowledge on which natural environments in rural areas work best as supportive environments in rehabilitation.

### 4.1. Limitations

The intervention group and the control group will not be randomly recruited, and therefore we cannot safely determine whether they are completely comparable, even though we will match the participants regarding age, gender, and diagnosis. Instead of randomisation, we chose a method that can be described as a natural experiment or a quasi-experiment. This can be seen as a limitation, but by choosing this alternative, the risk of loss or change in the control group decreases due to dissatisfaction with not being able to belong to the intervention group, and at the same time, one can assume that the values are similar to those given in the ordinary rehabilitation. That is, this reduces threats to external validity, which often limits the value of a Randomised Controlled Trial (RCT). By using this type of design, we can involve common health centres, job centres, and general practitioners, thereby making the results more generalisable. In addition, the lack of randomisation often facilitates the recruitment of a larger proportion of eligible participants, further increasing generalisability.

### 4.2. Closure

If this study shows positive effects and significantly can improve life quality and reduce stress among men with stress, anxiety, depression, post-cancer and chronic cancer, COPD, cardiovascular disease, or diabetes type II, the ‘Wildman Programme’ has a potential to be implemented as a rehabilitation programme for men in municipalities in Denmark. Pilot projects prior to this study indicate that the NBMC approach used in the ‘Wildman Programme’ can also have beneficial effects on groups of women, groups consisting of both men and women, and on groups of young people. In a future perspective, it would be interesting to investigate the effect of the programme and the methods on new target groups.

## Figures and Tables

**Figure 1 ijerph-17-03368-f001:**
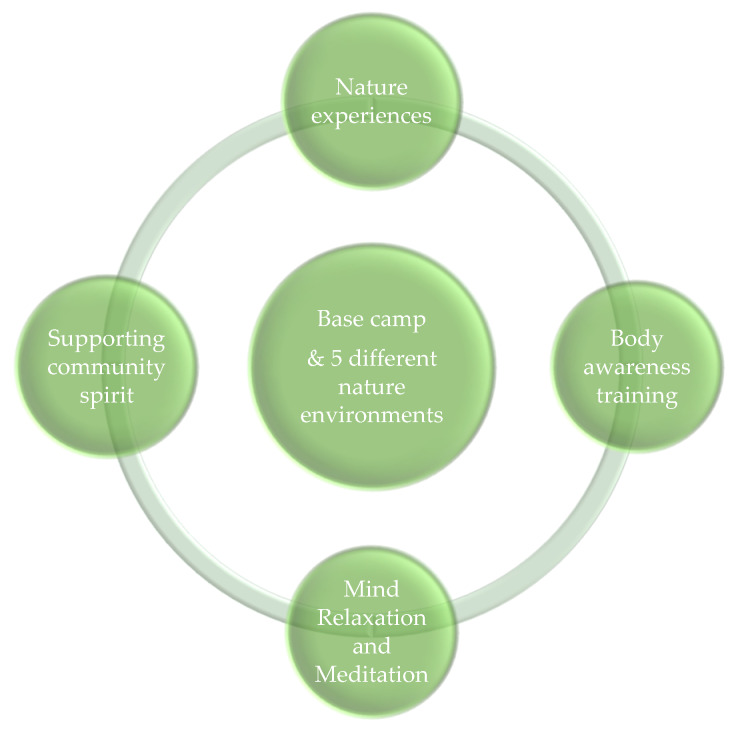
The elements in the Nature–Body–Mind–Community approach in the ‘Wildman Programme’.

**Figure 2 ijerph-17-03368-f002:**
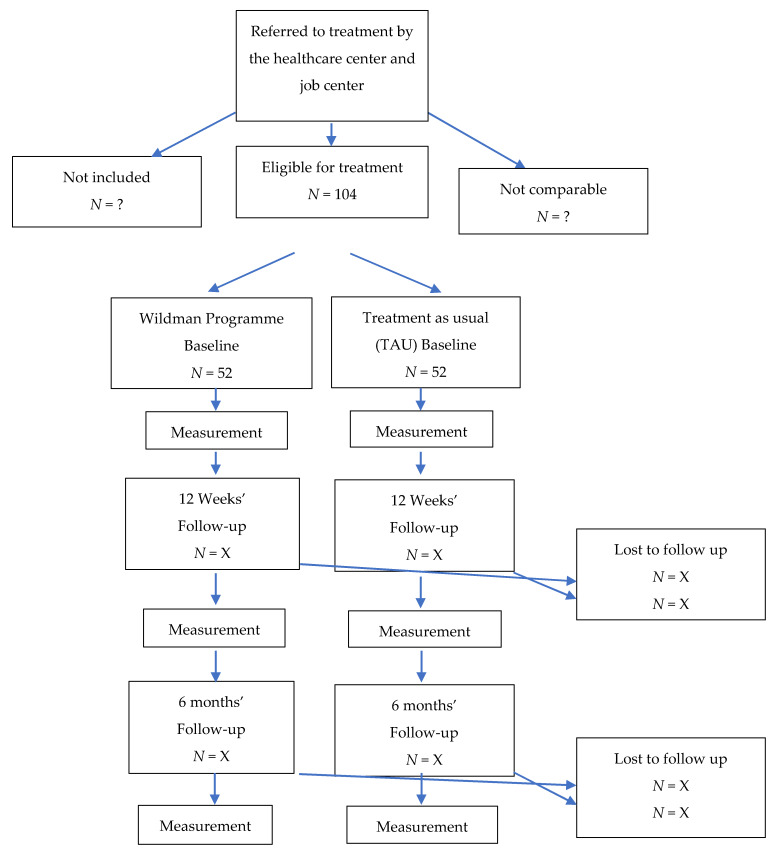
Flowchart for the study of the ‘Wildman Programme’.

**Table 1 ijerph-17-03368-t001:** The nature qualities of the five nature environments in the ‘Wildman Programme’.

Nature Environment	Nature Qualities
Meadowland	Open landscape, rich in species, overview but still closeness to nature
Hill	Overview of the landscape, greater perspective, observation of what is happening around you
Forest	Density, darkness, wandering shade, big trees, intense sensations from sounds, scents, and sight impressions
Tunnel valley with creek	Ice Age landscape, variation in nature, trees, sound of running water is amplified
Shore	Horizon, waves and currents, sand and stones, sun and wind

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
