# Peer review of "The Wildman Programme. A Nature-Based Rehabilitation Programme Enhancing Quality of Life for Men on Long-Term Sick Leave: Study Protocol for a Matched Controlled Study In Denmark"

_ijerph, 2020, doi:10.3390/ijerph17103368_

Round 1
Reviewer 1 Report
This study aims to examine the efficacy of Wildman Programme on male participants’ quality of life. But the manuscript is a semi-finished product. The writing logic is problematic and it doesn’t even include the results. Suggestions are listed for authors’ consideration.
- Please include the study results or finding in the end of abstract (line 23-24).
- Line 62-130. The introduction of “Wildman Programme” here and its theoretical background are not well written. Authors didn’t articulate closely how these theories apply to the programme. I suggest rephrase these paragraphs, please illustrate what is “Wildman Programme” first; and explain why and how these theories help the development of programme.
- Line 134-135. You didn’t study the “further aim” in current study.
- Line 136-144. The statements of research hypothesis, they did not look like research hypothesis. Please rephrase concisely.
- Line 153-156. Please explain how are the five nature environments selected and their attributes in PSD?
- Line 162. What is ICD? Please explain these abbreviations in the manuscript.
- Line 175, 176. will?! Tense.
- Line 193-220. The description should move to introduction and rephrase concisely. The intervention should include (1) treatments or activities to both control and experiment groups, and (2) differences across both groups.
- Line 221. Another 2.3.?!
- Line 222-275. Please have them concisely.
- Line 298. This section should be research procedure. Please cut out the redundancy and illustrate the procedure precisely.
- Line 243. “According to population standards for men (citation?)…high global quality of life.”
- Where are the results? And I can’t review the discussion without results.
Author Response
Reviewer 1
Reviewer: This study aims to examine the efficacy of Wildman Programme on male participants’ quality of life. But the manuscript is a semi-finished product. The writing logic is problematic and it doesn’t even include the results. Suggestions are listed for authors’ consideration.
Answer: Thank you for your review of this article! Your opinions have been valuable. This is a study protocol article, where the intention is to report how a study is intended to be conducted. Through a review process, the study's design and focus can be improved.
Reviewer: 1. Please include the study results or finding in the end of abstract (line 23-24).
Answer: Line 14-27. As stated above: this is a study protocol article, where the intention is to report how a study is intended to be conducted. The results will therefore be presented when the study is completed. To clarify this, we have rewritten our Abstract to make it clear that this is a study protocol. First, the second sentence now begins with: "This study protocol aims to describe the 'Wildman Program'…". Second, we have rewritten Abstract so that the sentences sometimes are expressed in the future perfect instead of present, so that it is consistent with what has not yet been implemented. And throughout the article, we have also changed tense, so that it is apparent that this study is not yet completed.
Reviewer: 2. Line 62-130. The introduction of “Wildman Programme” here and its theoretical background are not well written. Authors didn’t articulate closely how these theories apply to the programme. I suggest rephrase these paragraphs, please illustrate what is “Wildman Programme” first; and explain why and how these theories help the development of programme.
Answer: Line 40-132 and 134-165. We have rewritten this according to your and other reviewers' suggestions.
Reviewer: 3. Line 134-135. You didn’t study the “further aim” in current study.
Answer: Line 166-173. Since this is a study protocol, the aim is to accomplish that.
Reviewer: 4. Line 136-144. The statements of research hypothesis, they did not look like research hypothesis. Please rephrase concisely.
Answer: Line 171-173. We have now clarified that: “nature-based Wildman rehabilitation is suitable for men” is our conceptual working hypothesis.
Reviewer: 5. Line 153-156. Please explain how are the five nature environments selected and their attributes in PSD?
Answer: Line 196-206: We have clarified our selection of natural environments by the following phrases: The nature environments are selected with inspiration from the perceived sensory dimensions (PSD): Serene nature, rich in species and refuge which are identified as most important to people with symptoms of stress [33, 50, 51]. The five local nature environments (see table 1) have been selected during the pilot project [49]. They are based on feedback and preferences from former participants. The former pilot study shows that the participants have preferences for different type of nature, therefore both shore, forest, tunnel valley, hill and meadowland are represented in the ‘Wildman Programme’. These natural environments are carefully selected but not unusual or exceptional, rather they represent a spectrum of natural environments in Denmark. The nature environments are also selected so that the participants easily can access them by private or public transport and the environments are easy for the participants to visit alone or with their families after the course has ended.
Reviewer: 6. Line 162. What is ICD? Please explain these abbreviations in the manuscript.
Answer: Line 210-211: We have now clarified that ICD is the International Classification of Diseases according to WHO (2018).
Reviewer: 7. Line 175, 176. will?! Tense.
Answer: Yes, that is correct: will. However, throughout the article, we have now changed tense, so that it is apparent that this study is not yet completed.
Reviewer: 8. Line 193-220. The description should move to introduction and rephrase concisely. The intervention should include (1) treatments or activities to both control and experiment groups, and (2) differences across both groups.
Answer: Line 31-131, 134-165 and 207-241. The introduction has now been rewritten to coincide with several reviewers. This paragraph describes the intervention itself. The section "Participants", describes what the control group receives and how their rehabilitation differs from the intervention group.
Reviewer: 9. Line 221. Another 2.3.?!
Answer: Thank you for making us aware of this. It is of course 2.4, and we have changed the numbering below.
Reviewer: 10. Line 222-275. Please have them concisely.
Answer: Line 271-336. The paragraph is now rewritten with more references.
Reviewer: 11. Line 298. This section should be research procedure. Please cut out the redundancy and illustrate the procedure precisely.
Answer: Line 359. Since this is a study protocol article, this is the correct heading of the section.
Reviewer: 12. Line 343. “According to population standards for men (citation?)…high global quality of life.”
Answer: Line 404. A reference has been added:
Mizobuchi, Hideyuki. "Measuring world better life frontier: a composite indicator for OECD better life index." Social Indicators Research 118.3 (2014): 987-1007.
Reviewer: 13. Where are the results? And I can’t review the discussion without results.
Answer: As this is a study protocol article, the results will be presented in a follow-up article.
Reviewer 2 Report
Thank you for the opportunity to read and review this interesting manuscript. The article discusses the development of a programme of nature-based therapy for men in Denmark. As the authors themselves note in the discussion, this specific focus on men is a unique approach, and a demographic category often overlooked in research on nature/eco-therapies, and thus is it offers a potentially fantastic original contribution to knowledge. Whilst I have enjoyed reading the article, and am always pleased to learn about the development of new approaches in this area, I have some comments and concerns, which I hope, might be useful for the authors to reflect on.
Firstly, the abstract needs to make clear that this is a protocol, rather than a study. Indeed, an explanation in the paper itself that this is a protocol would be useful. It isn’t until over half way through that this is made clear. Indeed, I’m not quite sure what the purpose of this piece is. It feels as though it would be better to wait until there is empirical data to present, and write this up as a full study. The materials and methods are not detailed enough (understandably, given the subjective, place-based, individual nature of the intervention – and a challenge across all evaluations of nature-based therapies) to be easily replicated, so a ‘protocol’ seems an odd-choice here.
The introduction makes specific contextual reference to a Danish setting. This specific Danish focus is absent from both the title and abstract. It would be useful to incorporate this somehow. Or, possibly, open with a broader global contextualisation before moving to the specifics of Denmark.
Given that ‘nature’ in Japan is likely very different than ‘nature’ in Denmark, I did not feel the comparison to Shinrin-Yoku was useful here. There is a lot of literature discussing ‘green care’ in a European context that may be more appropriate, otherwise the authors risk positioning ‘nature’ as some form of universal essential.
A detailed explanation/definition of what the authors mean by ‘nature’ would be helpful. Are we talking about isolated wilderness, or urban parks? This is probably one of the biggest challenges of this paper. ‘Nature’ is never defined, or conceptualised. Yet it is a highly differentiated and contested term. Given it is so important and doing so much work in this article and research, it needs to be unpacked in a lot more detail.
The article presents multiple theoretical frameworks. However, no rationale for the choices of these is given. Why these? Why so many? Why not others (Gesler’s ‘therapeutic landscapes’ for example)?
I’m a little uncomfortable with the use of ‘natural environments’ as some sort of binary to ‘urban’. There is a risk of romanticising the rural – which has its own set of health problems and stressors – as well as suggesting that there is a ‘natural’ environment out there, given we live in what has been termed the Anthropocene, with landscapes having been managed and modified by humanity for thousands of years.
The paper regularly makes over-the-top, unscientific generalisations and unsupported claims. All of this results in a 2-dimensional, rose-tinted characterisation of ‘nature’. For example: “In nature, however, most of the things you experience are of a soft character.”; “Nature without threats automatically causes positive feelings” – automatically? What of people who are disgusted by bugs, vermin? A pool of reeking stagnant water? Cow-dung? Dead birds? Thorn-bushes?; “In natural environments, compared to urban environments, it is easier for us to judge when a threat is over” – in ALL ‘natural environments’?
What do the authors mean when they say that “many people today need to rebuild their contact with nature” – there are trees outside of my office window, is this ‘contact with nature’?
The authors discuss “sensory experiences in nature” – what makes these sensory experiences different/better than sensory experiences in an urban landscape? The pleasing aroma of coffee or fresh bread?
The authors discuss how “Being together in nature can strengthen positive relationships because social barriers and hierarchy are removed, and the participants are more open to others in the group.” – what is specific about ‘nature’ that does this? Why does nature have some inherent power to ‘strengthen positive relationships’? Social barriers and hierarchy can be removed in ‘urban’ settings too. What makes this about ‘nature’ rather than simply the group-culture and an experienced facilitator? Could the same ‘team spirit’ not also be created in an urban-activities programme?
There is an over-reliance on evolutionary claims and attributions, rather than empirical accounts from participants, or grounding in other literature related to nature-based/eco-therapies (of which there is much).
There is no detail at all about ethical approval. This needs to be included in detail.
The reason to exclude people who have a ‘lack of motivation’ needs further explanation and rationale, particularly given that people affected by depression are aiming to be recruited. How do these categories interact? It seems to potential bias the study.
Relatedly, ‘drop-out’ is only conceptualised as being due to health, work, or ‘lack of motivation’. Lack of motivation here is covering a lot of things, and places the blame solely on participants. It doesn’t appear to have been considered that participants could perhaps, not enjoy the programme, or find it unhelpful.
Limitations are not discussed at all.
The claim is made in the discussion that “In Denmark, we still do not have the same practice to integrate nature in health programmes” – given the growing ‘Green Care’ & care-farming sector in Denmark (Thodberg, K, “Status of Green Care in Denmark”, and see also https://groenomsorg.dk/), this seems a little bit of an over-the-top claim.
There is also a tension between the claimed exceptionalness of the ‘natures’ that people are taken to, and the idea that this programme “has a potential to be implemented for men in municipalities all over Denmark”. How can this be controlled for?
The authors report no conflicts of interest. Do any of them have any involvement in the establishment/development of the Wildman Programme that this paper seeks to evaluate?
Whilst it is pleasing to see further developments of nature-based therapies for groups whom are often overlooked by this sector, the enthusiasm for this programme needs to be tempered by a much more nuanced and careful evaluation and analysis that approaches ideas more critically and unpacks complexity in greater detail.
Author Response
Reviewer 2
Reviewer: Thank you for the opportunity to read and review this interesting manuscript. The article discusses the development of a programme of nature-based therapy for men in Denmark. As the authors themselves note in the discussion, this specific focus on men is a unique approach, and a demographic category often overlooked in research on nature/eco-therapies, and thus is it offers a potentially fantastic original contribution to knowledge. Whilst I have enjoyed reading the article, and am always pleased to learn about the development of new approaches in this area, I have some comments and concerns, which I hope, might be useful for the authors to reflect on.
Answer: Thank you and thank you for your thorough review of this article! Your opinions have been valuable.
Reviewer: Firstly, the abstract needs to make clear that this is a protocol, rather than a study. Indeed, an explanation in the paper itself that this is a protocol would be useful. It isn’t until over half way through that this is made clear. Indeed, I’m not quite sure what the purpose of this piece is. It feels as though it would be better to wait until there is empirical data to present, and write this up as a full study. The materials and methods are not detailed enough (understandably, given the subjective, place-based, individual nature of the intervention – and a challenge across all evaluations of nature-based therapies) to be easily replicated, so a ‘protocol’ seems an odd-choice here.
Answer: Line 4, line 14-27. The title contains "study protocol" as a sub-heading, but we understand that it is possible to miss this information. We have rewritten our Abstract to make it clear that this is a study protocol. First, the second sentence now begins with: "This study protocol aims to describe the 'Wildman Program'…". Second, we have rewritten Abstract so that the sentences sometimes are expressed in the future perfect instead of present, so that it is consistent with what has not yet been implemented. And throughout the article, we have also changed tense, so that it is apparent that this study is not yet completed. The purpose of this study protocol is partly to inform about the study, but first and foremost to report how the study is intended to be conducted. Through the review process, the design and focus of the study can be improved.
Reviewer: The introduction makes specific contextual reference to a Danish setting. This specific Danish focus is absent from both the title and abstract. It would be useful to incorporate this somehow. Or, possibly, open with a broader global contextualization before moving to the specifics of Denmark.
Answer: Line 5. The Abstract ends with: “The results of this study will be important to state whether the method in the ‘Wildman Programme’ can be implemented as a rehabilitation offer in the Danish Healthcare System to help men with different health problems.” However, to clarify that the study intends to be conducted in Denmark, we have added this in the title of the manuscript.
Reviewer: Given that ‘nature’ in Japan is likely very different than ‘nature’ in Denmark, I did not feel the comparison to Shinrin-Yoku was useful here. There is a lot of literature discussing ‘green care’ in a European context that may be more appropriate, otherwise the authors risk positioning ‘nature’ as some form of universal essential.
Answer: Line 55-58. We have deleted that paragraph and are writing the following instead: A large number of scientific studies suggest that forest therapy is an excellent intervention in reducing stress levels and depression in adults. However, most studies are not sufficiently well accomplished regarding control groups and accuracy.” In addition, we have added two systematic reviews regarding how forest therapy can relieve stress and depression.
Reviewer: A detailed explanation/definition of what the authors mean by ‘nature’ would be helpful. Are we talking about isolated wilderness, or urban parks? This is probably one of the biggest challenges of this paper. ‘Nature’ is never defined, or conceptualised. Yet it is a highly differentiated and contested term. Given it is so important and doing so much work in this article and research, it needs to be unpacked in a lot more detail.
Answer: Line 142 and 187-189 and 200-206. We have rewritten the entire introduction where we clarify what kind of natural environments we are talking about. With regard to the "Wildman Programme", we write in the Introduction that these environments are located in rural areas, and in the Venue paragraph we write the following: “The intervention programme takes place in five different natural environments in the countryside. They are not located far from urban areas, but participants in a previous pilot study have felt that they are completely surrounded by natural environments.”
Reviewer: The article presents multiple theoretical frameworks. However, no rationale for the choices of these is given. Why these? Why so many? Why not others (Gesler’s ‘therapeutic landscapes’ for example)?
Answer: Line 31-165. Thanks for pointing this out. There is a clear connection between how the theories are interconnected and why they are selected. We have rewritten the entire introduction to clarify this.
Reviewer: I’m a little uncomfortable with the use of ‘natural environments’ as some sort of binary to ‘urban’. There is a risk of romanticising the rural – which has its own set of health problems and stressors – as well as suggesting that there is a ‘natural’ environment out there, given we live in what has been termed the Anthropocene, with landscapes having been managed and modified by humanity for thousands of years.
Answer: Line 46-50 and 59-62 and 95-97 and 187-189 and 196-200. We have rewritten the introduction to clarify how nature areas can be perceived, and throughout the article we relate to this introduction. In addition, we highlight that the research on perceived sensory dimensions (PSD) is used in the selection of case study areas.
Reviewer: The paper regularly makes over-the-top, unscientific generalisations and unsupported claims. All of this results in a 2-dimensional, rose-tinted characterisation of ‘nature’. For example: “In nature, however, most of the things you experience are of a soft character.”; “Nature without threats automatically causes positive feelings” – automatically? What of people who are disgusted by bugs, vermin? A pool of reeking stagnant water? Cow-dung? Dead birds? Thorn-bushes?; “In natural environments, compared to urban environments, it is easier for us to judge when a threat is over” – in ALL ‘natural environments’?
Answer: Line 46-50 and 95-96 and 90-91 and 95-97. As stated above, we have rewritten the introduction to clarify how nature areas can be perceived. We only describe what people can discover with their senses. Cow dung, storms and stagnant water are not perceived as particularly pleasant and rose-tinted, but they are clear features that are easy to relate to. The sentence: “In nature, however, most of the things you experience are of a soft character” exemplifies all the research related to the ART theory that has been conducted over thirty years. And the sentence “Nature without threats automatically causes positive feelings” exemplifies all the research related to the AAT theory that has been conducted over thirty years. We have added a comprehensive systematic review ? to each of these two theories to clarify this. We add this paragraph in the introduction: “Both theories, AAT and ART, are based on evolutionary theory. They claim that not all natural areas are restorative. The restorative natural environments should contain special characteristics that most people interpret as safe and restful, and the theories involve a number of such qualities.” Line 94 and 97.
Reviewer: What do the authors mean when they say that “many people today need to rebuild their contact with nature” – there are trees outside of my office window, is this ‘contact with nature’?
Answer: We believe that the introduction now can lead the reader to what this is about, but adds a reference that can further elucidate what it is about: Walter, P. 2013. Greening the Net Generation: Outdoor Adult Learning in the Digital Age. Adult Learning, 24(4): 151-158. https://doi.org/10.1177/1045159513499551
Reviewer: The authors discuss “sensory experiences in nature” – what makes these sensory experiences different/better than sensory experiences in an urban landscape? The pleasing aroma of coffee or fresh bread?
Answer: Line 131-132. In the Introduction, we clarify this at the end of 1.1 Rehabilitation in natural environments. The AAT argues that humans have easier to interpret and value sensory experiences from nature, and ecotherapy claims that sensory experiences in nature can activate the parasympathetic nervous system which might help humans to restore the body and mind.
Reviewer: The authors discuss how “Being together in nature can strengthen positive relationships because social barriers and hierarchy are removed, and the participants are more open to others in the group.” – what is specific about ‘nature’ that does this? Why does nature have some inherent power to ‘strengthen positive relationships’? Social barriers and hierarchy can be removed in ‘urban’ settings too. What makes this about ‘nature’ rather than simply the group-culture and an experienced facilitator? Could the same ‘team spirit’ not also be created in an urban-activities programme?
Answer: Line 319-323. Of course, this can be achieved in urban areas. However, there is a great deal of research suggesting this. We have added the following sentence: “Research indicate that activities in natural areas often become more spontaneous, due to the fact that the environment is more unaffected, far less rigid, strict and organized as in built environments. This in turn leads to participants in meetings get more informal contact areas, which in turn leads to opportunities for spontaneous meetings and networks” followed by a reference.
Reviewer: There is an over-reliance on evolutionary claims and attributions, rather than empirical accounts from participants, or grounding in other literature related to nature-based/eco-therapies (of which there is much).
Answer: Line 40-165. Since this is a study protocol we will return with empirical material. The model we propose, will of course test the evolutionary theories that are strong in the "nature and health" field of research: AAT, ART, SET and ecotherapy are all strongly linked to evolutionary hypotheses, as well as other environmental psychological theories, such as the biophilia hypothesis, the savannah theory, the prospect-refuge theory and others.
Reviewer: There is no detail at all about ethical approval. This needs to be included in detail.
Answer: Line 184-185. Thank you for making us aware of this! It is now included. Participation in the ‘Wildman Programme’ is voluntary, and the study has been ethical approved by the University of Southern Denmark Research & Innovation Organization, SDU RIO.
Reviewer: The reason to exclude people who have a ‘lack of motivation’ needs further explanation and rationale, particularly given that people affected by depression are aiming to be recruited. How do these categories interact? It seems to potential bias the study.
Answer: Thanks for your comment on this. We have decided to take ‘lack of motivation’ out of the study as an exclusions criterium. Participation in the study is voluntary.
Reviewer: Relatedly, ‘drop-out’ is only conceptualized as being due to health, work, or ‘lack of motivation’. Lack of motivation here is covering a lot of things, and places the blame solely on participants. It doesn’t appear to have been considered that participants could perhaps, not enjoy the programme, or find it unhelpful.
Answer: Line 394. Good point. We have chosen to change this into ‘lack of interest in the course or doesn’t get enough out of the course’.
Reviewer: Limitations are not discussed at all.
Answer: Line 450-464. We have added this paragraph.
Reviewer: The claim is made in the discussion that “In Denmark, we still do not have the same practice to integrate nature in health programmes” – given the growing ‘Green Care’ & care-farming sector in Denmark (Thodberg, K, “Status of Green Care in Denmark”, and see also https://groenomsorg.dk/), this seems a little bit of an over-the-top claim.
Answer: Line 425-427. We have modified the paragraph by adding ‘on a strategic level’ and by adding: “as in e.g. Sweden, Norway and the Netherlands” and two references.
Reviewer: There is also a tension between the claimed exceptionalness of the ‘natures’ that people are taken to, and the idea that this programme “has a potential to be implemented for men in municipalities all over Denmark”. How can this be controlled for?
Answer: Line 200-206. The nature environments which are chosen in this study are not exceptional in Denmark, on the contrary. However, the cases are carefully selected so that they contain different restorative natural values. This is now made clear under the heading "Venue".
Reviewer: The authors report no conflicts of interest. Do any of them have any involvement in the establishment/development of the Wildman Programme that this paper seeks to evaluate?
Answer: Line 487-491.This phrase has been added: Simon Høegmark has been a part of the development of the ‘Wildman Programme’. However, the ‘Wildman Programme’ will during the study be led by staff from the health centers in the Municipality of Svendborg and Faaborg-Midtfyn. Simon owns a small company and he and his companion have educated the staff in the methods of the ‘Wildman Programme’. Simon Høegmark gets no financial benefits from the project.
Whilst it is pleasing to see further developments of nature-based therapies for groups whom are often overlooked by this sector, the enthusiasm for this programme needs to be tempered by a much more nuanced and careful evaluation and analysis that approaches ideas more critically and unpacks complexity in greater detail.
Answer: Thanks for making us aware of this. We have moderated the text according to your comment.
Round 2
Reviewer 1 Report
- Line 62, does it really have a theory call “AAT”? I didn’t found the term “AAT” at least in the reference your cited [22]. It sounds like SRT (stress reduction theory), line 65-75.
- Line 156-158, that is not a hypothesis. Suggesting delete it.
- Line 171-179, the section (1.3. Aim) is for illustrating research purposes, suggesting don’t mix research hypotheses here. And if research hypotheses really need to be articulate, please add a section “research hypotheses” and use a format correctly, such as,
H1: participants receiving the intervention has a significant improvement in their quality of life measured by the WHOQOL-BREF scale.
This is a simple investigation for examining the efficacy of Wildman Programme, so just explain what’s your purpose clearly and/or anticipate outcomes.
- Line 213-215, no need mention hypothesis again here.
- Line 218-238, these paragraphs are more like study design and repeating in line 359-399. I suggest rename “3. Status of the Project” to “Research Design” and avoid a repeat.
Author Response
Reviewer number 1.
Author: Thank you for taking the time to convey valuable views on the manuscript
Reviewer: 1. Line 62, does it really have a theory call “AAT”? I didn’t found the term “AAT” at least in the reference your cited [22]. It sounds like SRT (stress reduction theory), line 65-75.
Answer: Line 94. We have changed to SRT - Stress Reduction Theory throughout the manuscript.
Reviewer: 2. Line 156-158, that is not a hypothesis. Suggesting delete it.
Answer: Line 202. This has been deleted.
Reviewer: 3. Line 171-179, the section (1.3. Aim) is for illustrating research purposes, suggesting don’t mix research hypotheses here. And if research hypotheses really need to be articulate, please add a section “research hypotheses” and use a format correctly, such as,
H1: participants receiving the intervention has a significant improvement in their quality of life measured by the WHOQOL-BREF scale.
This is a simple investigation for examining the efficacy of Wildman Programme, so just explain what’s your purpose clearly and/or anticipate outcomes.
Answer: Line 222-227. We have changed according to your proposals, so that we now delete hypotheses under the heading aim. Instead, we highlight what expectations we have of the project.
Reviewer: 4. Line 213-215, no need mention hypothesis again here.
Answer: Line 263. We have changed this to “We expect that…”
Reviewer: 5. Line 218-238, these paragraphs are more like study design and repeating in line 359-399. I suggest rename “3. Status of the Project” to “Research Design” and avoid a repeat.
Answer: Line 410- 441. We have made new headline and avoid a repeat.
Reviewer 2 Report
Thank you for the opportunity to see this again. I’m impressed by the authors evident work on this, and their efforts to engage with the reviewers comments, particularly at a time when there is so much chaos in the world! The authors have done a lot of work, and refined the article into something stronger. However, I still have some concerns.
Whilst the authors have introduced more clarity in the title and abstract about this being a protocol, I still feel this choice needs to be explained and justified within the body of the manuscript itself – the word protocol is never mentioned in the paper after the abstract. Given that this being a protocol is the rationale for responding to some of the other reviewer's queries, the reasoning for producing a protocol rather than waiting for empirical results needs to be addressed within the manuscript itself. The authors explain this rationale in their response to reviewers, but this should be incorporated into the paper itself. The tense change throughout makes a huge difference, and is a very useful edit – thank you.
One of my major problems with the previous draft of this manuscript was that the manuscript lacked definition, explanation, and conceptualisation of what the authors meant by ‘nature’. Unfortunately I do not feel this has been adequately remedied. The authors have leaned further into the idea of ‘natural environments’, another phrase which I pointed out was in part problematically contested. What is ‘natural’? What is ‘nature’? These terms need to be addressed. Hinchliffe’s ‘Geographies of Nature’ may be helpful here. The authors need to clearly explain what they mean when they speak of ‘natural environments’. I feel this is particularly important given the interdisciplinary leanings of this journal. There is a much higher need to be clear and specific about how different phrases/concepts are used and mobilised, given that there are different interpretations and connotations.
I did not feel the turn to ancient literature at the start was helpful, and didn’t seem to fit with the rest of the paper’s grounding in evolutionary theories. The reference to Gilgamesh for example, mentions gardens. Many urban spaces have gardens. Gardens are managed spaces, man-made, certainly far from ‘natural’, so this confused the message further, rather than clarifying. Do the authors mean ‘green-spaces’ ?
I was pleased to see the authors recognise that “not all natural areas are restorative”. However, this doesn’t seem to then connect up with any discussions later on in the paper. How does this link to, and inform, what is presented in the sections under 2.4?
The authors suggest that their introduction now explains what a connection to/with nature is. I’m afraid I’m still confused as to what the authors mean by this and how it is being used for their argument. Possibly this idea may be a core part of the authors discipline and they do feel it needs to be defined, however, again, given the papers potential appeal to an interdisciplinary audience, this needs to be exemplified further to give the precise meaning the authors desire. May I recommend Tam (2013) Concepts and measures related to connection to nature: Similarities and differences. This piece highlights the multiple meanings and interpretations attached to the concept of ‘connection to nature’.
The authors place a lot of faith and responsibility for their claims on the ART literature. Whilst this is fine, it may be worth recognising that there are critiques of this framework. For example Yoye & Dewitte’s ‘Nature's broken path to restoration. A critical look at Attention Restoration Theory’.
Again, there is the creep of unsubstantiated phrases and generalisations such as “Throughout history, men have been attracted to activities in nature, and it is still that way today”. This does not feel appropriate for an academic article, without being evidenced or explored in more complexity. It perceives ‘men’ as a homogenous category, unconnected to other intersectional areas, for example, disability, race, class. The idea that all men are tempted by ‘laborious challenges such as long hikes’ is unhelpful hyperbole.
Author Response
Reviewer number 2.
Reviewer: Thank you for the opportunity to see this again. I’m impressed by the authors evident work on this, and their efforts to engage with the reviewers comments, particularly at a time when there is so much chaos in the world! The authors have done a lot of work, and refined the article into something stronger.
Answer: Thank you, and thanks for taking the time to convey valuable views on the manuscript
Reviewer: 1. However, I still have some concerns. Whilst the authors have introduced more clarity in the title and abstract about this being a protocol, I still feel this choice needs to be explained and justified within the body of the manuscript itself – the word protocol is never mentioned in the paper after the abstract. Given that this being a protocol is the rationale for responding to some of the other reviewer's queries, the reasoning for producing a protocol rather than waiting for empirical results needs to be addressed within the manuscript itself. The authors explain this rationale in their response to reviewers, but this should be incorporated into the paper itself. The tense change throughout makes a huge difference, and is a very useful edit – thank you.
Answer: Line 219- 221. We find it appropriate to add this information under Aims, and have now changed section 1.3 Aim with an amendment describing that this is a study protocol and why we want to present it: The purpose of this study protocol is to inform about the study and how it is intended to be conducted. Moreover, through the review process, the design and focus of the study can be improved.
Reviewer: 2. One of my major problems with the previous draft of this manuscript was that the manuscript lacked definition, explanation, and conceptualisation of what the authors meant by ‘nature’. Unfortunately I do not feel this has been adequately remedied. The authors have leaned further into the idea of ‘natural environments’, another phrase which I pointed out was in part problematically contested. What is ‘natural’? What is ‘nature’? These terms need to be addressed. Hinchliffe’s ‘Geographies of Nature’ may be helpful here. The authors need to clearly explain what they mean when they speak of ‘natural environments’. I feel this is particularly important given the interdisciplinary leanings of this journal. There is a much higher need to be clear and specific about how different phrases/concepts are used and mobilised, given that there are different interpretations and connotations. I did not feel the turn to ancient literature at the start was helpful, and didn’t seem to fit with the rest of the paper’s grounding in evolutionary theories. The reference to Gilgamesh for example, mentions gardens. Many urban spaces have gardens. Gardens are managed spaces, man-made, certainly far from ‘natural’, so this confused the message further, rather than clarifying. Do the authors mean ‘green-spaces’?
Answer: Line 51-82. This is a question that requires some space to answer properly. The definition of nature has been developed in landscape architecture and environmental psychology, and is more or less used as an umbrella term in the field of nature and health. The parts of this nature that are experienced as recreational and restorative by our participants are interesting. One tool used in this study is the Perceived Sensory Perception - PSD. On page two there is now a paragraph describing this.
Reviewer: 3. I was pleased to see the authors recognise that “not all natural areas are restorative”. However, this doesn’t seem to then connect up with any discussions later on in the paper. How does this link to, and inform, what is presented in the sections under 2.4?
Answer: Line 323-326. Section 2.4 begins with the following: “In the ‘Wildman Programme’ the participants will be introduced to different nature environments that are expected to have a supportive effect. The environments are selected based on nature qualities that according to the perceived sensory dimensions can make the participants feel comfortable and inspire to inner peace.” The natural environments are thus chosen to be attractive and restorative. If they do not work as intended, we will receive feedback on this from the participants and on the quantitative results.
Reviewer: 4. The authors suggest that their introduction now explains what a connection to/with nature is. I’m afraid I’m still confused as to what the authors mean by this and how it is being used for their argument. Possibly this idea may be a core part of the authors discipline and they do feel it needs to be defined, however, again, given the papers potential appeal to an interdisciplinary audience, this needs to be exemplified further to give the precise meaning the authors desire. May I recommend Tam (2013) Concepts and measures related to connection to nature: Similarities and differences. This piece highlights the multiple meanings and interpretations attached to the concept of ‘connection to nature’.
Answer: Line 204-208. Thank you, it is a good overview that covers eco-psychology, Wilson's Biophilia hypothesis and include several well-known names in environmental psychology. We add this reference in the following paragraph (1.2), and supplement with a two reviews containing a number of central theories of nature connectedness and place attachment: “Nature connection, as we define it, includes place attachment. It is about the interaction between person, process and place and contains both cognitive, affective and behavioral components. The person in question must feel that the place and the activities it invites to, provides security, meaning and involve a flow of self-rewarding activities that can bestow a moment's happiness”.
Reviewer: 5. The authors place a lot of faith and responsibility for their claims on the ART literature. Whilst this is fine, it may be worth recognising that there are critiques of this framework. For example Yoye & Dewitte’s ‘Nature's broken path to restoration. A critical look at Attention Restoration Theory’.
Answer: Line 126-129. It is true that the theory has been criticized. We add a systematic review that shows that the theory basically works, and follows up with the criticism and the reference you propose: “The theory has been tested in many studies and has largely been found to work. However, there are criticisms of some assumptions, such as that the mechanism has an evolutionary basis and the definition of soft fascination”.
Reviewer: 6. Again, there is the creep of unsubstantiated phrases and generalisations such as “Throughout history, men have been attracted to activities in nature, and it is still that way today”. This does not feel appropriate for an academic article, without being evidenced or explored in more complexity. It perceives ‘men’ as a homogenous category, unconnected to other intersectional areas, for example, disability, race, class. The idea that all men are tempted by ‘laborious challenges such as long hikes’ is unhelpful hyperbole.
Answer: Line 179-183. We remove these sentences and write instead: “In various studies, men have shown a strong interest in the wilderness and activities in the wilderness, including multi-day trips with overnight stays. A North American study found that the difference between various ages and ethnic groups among men was relatively small. Interest has been relatively constant since the 1980s, and is not least high among men in Northern Europe.” We add references to these statements.